# Molecular Targets for the Diagnosis and Treatment of Pancreatic Cancer

**DOI:** 10.3390/ijms251910843

**Published:** 2024-10-09

**Authors:** Robert G. Pergolizzi, Steven T. Brower

**Affiliations:** 1Department of Vascular Surgery, Englewood Health, Englewood, NJ 07631, USA; robert.pergolizzi@ehmchealth.org; 2Department of Surgical Oncology and HPB Surgery, Englewood Health, Englewood, NJ 07631, USA

**Keywords:** PC (pancreatic cancer), driver mutations, miRNA

## Abstract

Pancreatic cancer is one of the most aggressive and lethal forms of cancer, with a five-year survival rate of less than 10%. Despite advances in treatment modalities, the prognosis for pancreatic cancer patients remains poor, highlighting the urgent need for innovative approaches for early diagnosis and targeted therapies. In recent years, there has been significant progress in understanding the molecular mechanisms underlying pancreatic cancer development and progression. This paper reviews the current knowledge of molecular targets for the diagnosis and treatment of pancreatic cancer.

## 1. Introduction

Pancreatic cancer (PC) is the seventh leading cause of cancer-related deaths worldwide, with an estimated 495,773 new cases and 466,003 deaths in 2020 [1]. In recent years, the incidence of PC has risen. It accounts for about 2% of all cancers and is responsible for 5% of cancer-related deaths [2]. Most patients exhibit no clear symptoms during the disease’s development, progressing to advanced metastatic stages where tumor cells become highly invasive. Early diagnosis is challenging [2,3], making PC one of the deadliest malignant tumors. The high mortality rate is attributed to the aggressive nature of the disease, late-stage diagnosis, and limited treatment options. Despite potential radical treatments, the 5-year survival rate for patients is only 2–9%, with most eventually relapsing [4]. Pancreatic ductal adenocarcinoma (PDAC) is the most common type of PC [2]. 

Risk factors include family history, pancreatitis, age, sex, diabetes, and a variety of other factors that have been implicated in the etiology of PC to varying degrees. However, there are currently no global standard screening programs for high-risk individuals. To improve prognosis, recent advancements in the risk factors, diagnosis, and treatment of PC have been reviewed elsewhere [5,6].

Over the past few years, the incidence of PC has increased. Most cases are diagnosed at an advanced stage when surgical resection is often not feasible, and current therapies, such as chemotherapy and radiation therapy, offer limited benefit. The early detection of PC is critical for improving outcomes. This paper reviews the current knowledge of molecular targets for the diagnosis and treatment of pancreatic cancer, with a focus on key signaling pathways and therapeutic strategies that may be employed to improve early diagnosis of PC, improve survival, and suggest possible novel targets for intervention and prevention.

PC has been the subject of much research and many attempts to improve outcomes. However, the following issues remain major stumbling blocks to the diagnosis and treatment of PC: (a) At the point of diagnosis, PC has already metastasized, and very few treatment options are available to patients. (b) The aggressive nature of this cancer can be attributed to the substantial number of mutations acquired during its progression and its subsequent resistance to standard therapies such as chemotherapy and radiation. (c) The heterogeneity of the chemo-resistant subpopulations, particularly the tumor-initiating population known as cancer stem cells (CSCs), makes administrating conventional first-line treatments such as Gemcitabine more difficult. (d) Lastly, the tumor microenvironment and the early establishment of a metastatic niche by exosomes that facilitate the dissemination of cancer cells to distant organs also contribute to the incurability of this type of cancer [7].

## 2. Risk Factors for PC

PC primarily affects older adults but is increasingly observed in younger patients. The known risk factors include smoking, alcohol consumption, and chronic pancreatitis. Recent studies have also implicated abnormal metabolism, blood type, and glucose and lipid levels in pancreatic cancer development. Early diagnosis remains a challenge, as existing tumor markers lack specificity. Future research should focus on improving early detection methods to identify high-risk individuals and improve survival rates.

While age is a well-known risk factor, the role of sex in pancreatic cancer development, diagnosis, and treatment outcomes is an emerging area of interest. PC incidence and mortality rates are higher in men than in women. The frequency varies from 11.6 per 100,000 for men to 8.1 per 100,000 for women, with significant regional variations in these figures [8]. Understanding how sex influences pancreatic cancer can provide insights into the underlying mechanisms of the disease and potentially lead to sex-specific strategies for prevention and treatment. Several factors may contribute to this difference, including hormonal, genetic, and lifestyle factors. For example, men are more likely to smoke, which is a known risk factor for PC. About 25% of PC patients are or have been long-term smokers [8]. Additionally, differences in the expression of sex hormones and their receptors may play a role in the development of PC.

## 3. PC Biomarkers

Tumor biomarkers are molecules that can be detected in blood, tissue, or other bodily fluids and indicate the presence of cancer. The relative ease of blood monitoring and the non-invasive nature of the testing have made this a highly desirable field of study. Biomarker testing may help guide treatment decisions and improve outcomes. In pancreatic cancer, several biomarkers have been identified and used for diagnosis, prognosis, and monitoring of the disease. Some of the most important tumor biomarkers in PC are discussed here.

CA 19-9 and CEA: Carbohydrate antigen 19-9 (CA 19-9) is the most widely used biomarker for pancreatic cancer. Elevated levels of CA 19-9 in blood have been associated with pancreatic cancer, and it has been used for diagnosis, monitoring response to treatment, and detecting recurrence. Carcinoembryonic antigen (CEA) is another biomarker that may be elevated in pancreatic cancer. It is not as specific for pancreatic cancer as CA 19-9 but can be used in combination with other markers for monitoring disease progression. CA 19-9 and CEA are high molecular weight glycoproteins attached to the surface of tumor cells that have been used in the diagnosis and prognosis of gut-associated cancers. Unfortunately, due to their low sensitivity and specificity, they have not been useful in the diagnosis of other cancers [9].

Alterations in DNA methylation patterns are common in pancreatic cancer and can serve as biomarkers for the disease. DNA methylation biomarkers are being investigated for their potential use in early detection and monitoring of pancreatic cancer. Significant epigenetic changes have been identified in PC stem cells, particularly involving mutations in proteins that regulate chromatin and the proteins that control the epithelial–mesenchymal transition (EMT). These changes can occur in PC without changes in the DNA sequence in the stem cells, ultimately impacting the cell’s overall phenotypic state. Consequently, it may be possible that, even without targeting specific mutations, inhibiting epigenetic regulation processes can lead to the development of new PC therapies [10,11].

## 4. Genetic Factors in Pancreatic Cancer

Pancreatic cancer is a complex disease with a multifactorial etiology, influenced by both environmental and genetic factors. While environmental factors such as smoking and diet are known to contribute to pancreatic cancer risk, genetic factors also play a significant role. This section aims to summarize the current understanding of the influence of genetic factors on pancreatic cancer and explore potential implications for risk assessment, early detection, and treatment strategies.

Several genetic risk factors have been identified that increase the likelihood of developing pancreatic cancer. Chromosomal aberrations at loci 13q22.1, 15q14, 6p25.3, 12p11.21, 7q36.2, 21q21.3, 5p13.1, 21q22.3, 22q13.32, and 10q26.1 have been described in numerous PC cases [12,13,14,15]. PC-susceptible chromosomal deletions at 7p12, 1p36.33, 8q21.11, 17q12, and 18q21.32 have also been identified in a European GWAS study [16]. 

Recent studies have identified a clear familial basis for PC, with family history significantly increasing the risk of the disease [17,18]. The risk of familial PC rises exponentially with the number of first-degree relatives affected [3,19,20]. However, over 80% of PC cases result from sporadic mutations. Specific mutations in genes *KRAS, CDKN2A (P16), TP53, SMAD4, BRCA2, BRCA1, STK11, PRSS1,* and *MMR* are often observed in PC [4,21,22,23,24]. 

Several molecular biomarkers have been identified for the early diagnosis of PC, including CA 19-9, CEA, and KRAS mutations. CA 19-9 is the most widely used biomarker for PC, but its sensitivity and specificity are limited, particularly in the early stages of the disease. Other promising biomarkers, particularly microRNAs, are being investigated for their potential utility in early diagnosis and are discussed more fully in this review.

Mutations in genes such as *BRCA1, BRCA2, PALB2,* and *ATM* are associated with an increased risk of pancreatic cancer, particularly in individuals with a family history of the disease. These mutations are also linked to other cancers, such as breast and ovarian cancer, highlighting the importance of genetic testing and counseling for individuals with a family history of cancer. Several other biomarkers, such as *MUC1, SPANXN2,* and *LRG1,* have been studied in pancreatic cancer. These biomarkers are being investigated for their potential use in early detection, prognosis, and treatment response prediction. 

In addition to specific gene mutations, hereditary pancreatic cancer syndromes have been identified that increase the risk of developing pancreatic cancer. These syndromes, such as hereditary pancreatitis and familial atypical multiple mole melanoma (FAMMM) syndrome, are characterized by an increased risk of developing pancreatic cancer as well as other cancers [3]. Understanding these syndromes can help identify individuals at high risk of pancreatic cancer and implement appropriate screening and prevention strategies.

Genetic markers have been identified that may help detect pancreatic cancer at an early stage when treatment is more effective. For example, mutations in the *KRAS* gene and others outlined below are commonly found in pancreatic cancer and can be detected in the blood or pancreatic fluid of patients with the disease. Other genetic markers, such as alterations in DNA methylation patterns, are also being investigated for their potential role in early detection. The use of liquid biopsies to screen for mutations has become more widely studied as the sensitivity of detection has increased. Screening for *KRAS*, while very important for early detection, is not limited to PC alone, leaving an open question of which tumor is contributing to the detection. Recent work has suggested that the efficiency and sensitivity of predictive screening for PC are greatly enhanced when screening for other mutations in addition to *KRAS*, such as *TP53* [25]. 

The impact of some hypomethylated genes on PC is shown in Table 1 (adapted from Reshkin, ref. [26]). These have been shown to impact PC progression despite being unmutated. 

Genetic factors can also influence the response to treatment in patients with pancreatic cancer. For example, tumors with certain genetic mutations may respond differently to chemotherapy or targeted therapies. By identifying these genetic markers, clinicians can tailor treatment plans to individual patients, potentially improving outcomes and reducing side effects.

The genes commonly associated with pathogenic germline alterations are *KRAS, BRCA1, BRCA2, ATM, PALB2, MLH1, MSH2, MSH6, PMS2, CDKN2A,* and *TP53.* The *TP53* gene encodes a protein that helps regulate cell division and prevent tumor formation. Mutations in the *TP53* gene are found in a significant number of pancreatic cancers and are associated with a poorer prognosis. The *CDKN2A* gene encodes a protein that regulates cell division. Mutations in this gene are common in pancreatic cancer and can lead to uncontrolled cell growth. The Mismatch Repair Genes (*MLH1, MSH2, MSH6,* and *PMS2*) are involved in repairing errors that occur during DNA replication. Mutations in mismatch repair genes can lead to a condition called Lynch syndrome, which increases the risk of several cancers, including pancreatic cancer. Mutations in the *BRCA1* and *BRCA2* genes are well-known risk factors for breast and ovarian cancer, but they also increase the risk of pancreatic cancer. These mutations are more common in individuals with a family history of breast, ovarian, or pancreatic cancer. *PALB2* is a gene that is involved in repairing damaged DNA. Mutations in *PALB2* increase the risk of pancreatic cancer, particularly in individuals with a family history of the disease. Unfortunately, to date, only *BRCA1/2* and *PALB2* alterations are actionable. Platinum agents have shown some benefit in patients with *BRCA1/2-* or *PALB2* mutations in PC [8].

Numerous studies have revealed that KRAS mutations are major drivers of PC progression. Mutations in the *KRAS* gene are found in over 90% of pancreatic ductal adenocarcinomas [PDAC], the most common type of pancreatic cancer. *KRAS* mutations lead to uncontrolled cell growth and are early events in pancreatic cancer development. However, other mutations have been associated with the roughly 10% of PCs that lack *KRAS* mutations. Alternate potential drivers include gene fusions (*ALK, NRG1, NTRK, ROS1, FGFR2,* and *RET*), mutations (*BRAF, BRCA1/2, KRAS,* and *PALB2*), genetic amplifications *(HER2), MSI, dMMR,* or *TMB* [27]. This list is constantly being revised and updated. 

A recent study referred to four genes, “canonical driver genes in pancreatic adenocarcinoma” [28]. These are:Kirsten rat sarcoma (*KRAS*).Cyclin-dependent kinase inhibitor 2 (*CDKN2A/p16*).○aka MTS1 or multiple tumor suppressor 1.Tumor suppressor protein 53 (*TP53*).Small Mothers against Decapentaplegic homolog 4 (*MADH4/DPC4/SMAD4*).

There is considerable heterogeneity in the molecular profile of PC, but these genes are involved in a preponderance of cases, suggesting that targeting these altered genes and the pathways in which they act would provide logical therapeutic targets. Important research has shown that even the KRAS gene, long considered “untargetable”, is starting to yield to drugs. Sotorasib and Adagrasib, both approved by the FDA, effectively target the *KRAS* G12C mutation. However, although this mutation is common in lung cancer, the most common *KRAS* mutation in PC is G12D/V, against which these drugs have no effect [28]. These studies are providing a roadmap for the development of drugs against the PC *KRAS* variant. 

All members of the RAS family are activated by the binding of GTP, including *KRAS,* which is the most frequently observed mutation in PC [26]. This binding, and the subsequent release of GDP following hydrolysis, occurs with the assistance of guanine nucleotide exchange factors (GEFs). *KRAS* is inactivated by the hydrolysis of GTP, involving the cooperation of GTPase-activating proteins (GAPs). As illustrated in Figure 1, mutated *KRAS* cannot hydrolyze GTP to GDP, and consequently remains activated, continuing to signal pathways related to proliferation, nucleic acid and protein biosynthesis, and cell survival. It also plays a role in migration and invasion via connection with cellular integrins [26]. Mutated *KRAS* remains permanently activated. 

Another approach to silencing *KRAS* is the use of small interfering RNA (siRNA), which has shown considerable promise in in vitro studies. A small clinical study was performed and showed significant improvement in survival. However, there were only 15 patients and no controls [29]. It will be extremely interesting to see how this approach performs in a larger, controlled clinical study. 

A recent review has outlined the most common mutations in PC, their modes of action, and the drugs that have been and are being developed to subdue their effects [28]. Genetic factors clearly play a significant role in the development, diagnosis, and treatment of pancreatic cancer. Understanding these factors is essential for identifying individuals at high risk of the disease, implementing effective screening and prevention strategies, and developing personalized treatment approaches. Further research is needed to elucidate the complex interplay between genetic and environmental factors in pancreatic cancer and to translate these findings into clinical practice. Future studies must not rely on targeting a single oncogenic or mitotic pathway but must suppress the multiple stages of tumorigenesis of pancreatic cancer cells and their enabling microenvironment.

Mutations in tumor suppressor genes also have a powerful role in the progression of PCs. The *CDKN2A/p16* gene complexes are mutated in over 90% of PCs [30]. *TP53* and *MADH4/DPC4/SMAD4* mutations are seen in 60% and 50% of PCs, respectively, and appear late in tumor progression. The DNA repair genes *BRCA2* and *MLH1* were mutated in roughly 7% of PCs [30]. 

An ongoing project comparing gene mutations in PC to other hepatobiliary cancers at Englewood Health in New Jersey, USA, has demonstrated that the mutational burden differs between these cancers. This study has employed next-generation sequencing to determine specific changes among 648 genes associated with various types of cancer. For pancreatic cancers, the most common somatic mutations were *KRAS* (92%) and *TP53* (77%). In addition, the next most common mutations were *SMAD4* (31%), *CDKN2A* (23%), *CDKN2B* (15%), *ELF3* (13%), and *KMT2D* (7%). In contrast, the most common hepatobiliary mutations were *TP53* (63%), *KRAS* (50%), *SMAD4* (13%), *CDKN2A* (25%), *CDKN2B* (25%), and *KMT2D* (38%). The NGS of pancreatic cancer shows that *KRAS* and *TP53* mutations appear to occur at higher levels than hepatobiliary tumors (92% vs. 50% and 77% vs. 63%, respectively). *SMAD4* mutations were increased in pancreatic tumors over hepatobiliary tumors as well (34% vs. 13%). Most interestingly, *KMT2D* mutations occurred in 7% of pancreatic tumors compared to 38% of hepatobiliary tumors, perhaps signifying a unique genetic profile for hepatobiliary tumors, raising the possibility of determining the source of cancer in a liquid biopsy as the sensitivity of this approach continues to be refined.

## 5. Immunogenicity and Antigenicity of PC

“Tumor immunogenicity” has been defined as the ability to induce an immune response that can prevent or inhibit its growth [31]. The unique characteristics of the tumor microenvironment in PC results in the tumors being considered “immunologically cold”, and thereby not responding well to immunotherapy. Tumors with high immunogenicity usually have a high mutational burden and elevated expression of PD-L1. They also have high rates of T-cell infiltration, which is inhibited in PC by the dense tumor microenvironment [32]. 

Immunosuppressive mechanisms in tumors have been reviewed [32]. Immunosuppressive factors contributed by tumor cells include programmed cell death protein 1 (PD1) ligands, galectin 3, lactic acid, adenosine, prostaglandins, and transforming growth factor-β (TGFβ) [31]. Attempts have been made to overcome these inhibitors using pharmacological agents or antibodies. Other immunosuppressive mechanisms arising from non-tumor cells include regulatory T-cells that are attracted to some tumors by chemokines, and suppressor cells of myeloid lineage cells in which immunosuppression derives from nitrogen oxide (NO) and reactive oxygen species (ROS) [31].

Recent studies have suggested that pancreatic cancer creates a powerful immunosuppressive microenvironment [33], with low T-cell infiltration and a low rate of mutation, leading to immune tolerance in PC. The tumor microenvironment affects tumor growth and plays a role in drug therapy and resistance to immune checkpoint inhibitors (ICIs) [34]. These factors conspire to severely limit the immune response to PC, underscoring the need to quickly identify new therapeutic techniques for improving the prognosis of PC. This important area has been recently reviewed [35,36].

ICIs enhance antitumor immune responses by targeting receptors on the surface of T lymphocytes [37]. The PD-1/PD-L1 molecules and CTLA-4 immune checkpoints are being actively studied in the treatment of PC [38]. PD-L1 is expressed in PC cells and, by binding to PD-1, may inhibit T-cell activity, while the inhibition of CTLA-4 can activate T-cells and enhance the immune response against PC cells. However, the possible triggering of immunological side effects is one of the disadvantages of the PD-1/PD-L1 approach. With anti-CTLA-4, the immune response may be enhanced at first, which should prompt the immune system to attack cancer cells efficiently. However, the impact of anti-CTLA-4 is limited in some cancer types [36].

## 6. MicroRNAs in PC

MicroRNAs (miRNAs) are a class of non-coding RNAs that play important roles in regulating gene expression. In most cases, miRNA functions as a guide by base-pairing with target messenger RNA (mRNA) to negatively regulate gene expression [30]. Gene silencing can be performed either by degrading a specific mRNA or by inhibiting the translation of the transcript. miRNAs play vital roles in regulating metabolic and cellular pathways, including those controlling cell proliferation, differentiation, and survival [39,40,41,42,43,44,45,46]. Changes in the expression level of about 200 miRNAs have been observed in many types of cancer [47]. In one breast cancer study, patients’ plasma showed as many as 37 types of reduced expression miRNAs and 54 miRNAs over-expressed [48].

Some of these have been strongly associated with PC, especially miR-155, miR-21, miR-221, miR-222, miR-376a, and miR-301 [49,50,51,52,53,54,55]. Although there is still some uncertainty as to whether alterations seen in the levels of miRNA dysregulation in most tumors are a direct cause of the cancer or an indirect consequence of the changes in the cellular phenotype, monitoring levels of miRNAs in cancers will have diagnostic and prognostic value. Studies have demonstrated that decreased levels of certain miRNAs detected in the circulation appear to be a useful indicator of progression and evaluation of the aggressiveness of pancreatic cancer [49]. As shown in Table 2, numerous circulating miRNA molecules have demonstrated abnormal expression in pancreatic cancer.

Numerous studies have implicated miRNAs’ role in the pathogenesis of PC by influencing genetic changes in the expression of important genes such as *KRAS* and *TP53* and mediating changes in the tumor microenvironment that facilitate the growth and spread of tumor cells. miRNAs are important biomarkers for diagnosis, following response to treatment, prognosis, and therapeutic targets in PC. Much remains to be learned about their impact on normal cellular mechanisms and interactions in cancer, but levels of circulating miRNAs can reveal much about the growth and spread of PC.

## 7. miRNA in Diagnosis of PC

Because PC is often detected late with no proven mechanism for monitoring its development, and it has a propensity to aggressively metastasize, effective and highly detectable biomarkers would be extremely important in detecting the presence of the disease at its initial stage, allowing for early diagnosis and improved survival. Biomarkers that can be monitored in blood or other body fluids (“liquid biopsy”) would be highly desirable, and several candidates are currently under investigation [27]. Cells export apoptotic bodies containing miRNAs, shedding vesicles and exosomes into the bloodstream, enabling RNAs to survive in the presence of nucleases in serum. Using 8 miRs in one study (miR-125a-3p, miR-4294, miR-4476, miR-4530, miR-6075, miR-6799-5p, miR-6836-3p, and miR-6880-5p) achieved a sensitivity of 80.3%, a specificity of 97.6%, and an accuracy of 91.6% in the detection of pancreatic and biliary tract cancers compared to healthy controls, benign abnormalities, or other types of cancer [57]. This is a highly significant result. Furthermore, anecdotal reports indicate that specific miRNA levels can differentiate between PC and other pancreatic pathologies, although more work needs to be conducted to confirm these reports. 

Several miRNAs have been shown to have significantly increased levels of expression in PC in plasma when compared to healthy controls. These include miR-16, miR-21, miR-155, miR-181a, miR-181b, miR-196a, and miR-210, while let-7 miRNA expression is consistently lower in PC [58,59]. miRNA-21 and miR-221 expression has been shown to change with chemotherapy treatments, suggesting that the drug resistance of pancreatic cancer may result from miR-21 over-expression and inhibition of the activity of genes regulated by this molecule [59].

A serum biomarker that has been used to detect PC is CA 19-9, a cell surface glycoprotein complex commonly associated with pancreatic ductal adenocarcinoma. A high level of CA 19-9 is often an indication of PC, but other cancers and other, non-cancerous conditions, such as gallstones and cirrhosis, can also lead to elevated levels. Attempts are currently being made to determine the diagnostic efficacy of this indicator in conjunction with other molecules in the serum that are potentially useful in diagnosing cancer. One study measured CA 19-9 levels and several miRNAs in cancer, chronic inflammation, and normal volunteers. They determined that monitoring the levels of CA 19-9, miR-16, and miR-196a was indicative of early-stage PC [60].

One miRNA, miR-210, may potentially be very helpful in diagnosing PC. This miRNA is upregulated in hypoxia, which is often seen in PC. The molecular target of this miRNA is the SMAD-interacting protein [SIP1] gene that is often mutated in PC, which normally inhibits Cadherin E. It was suggested that hypermethylation of the SIP1 promoter may lead to abnormal expression of miR-210, which can be easily detected in serum, thus serving as a surrogate marker [56].

Another potentially useful miRNA for PC surveillance may be miR-150. Reduced levels of miR-150 appear to be correlated with the increased expression of the cell surface-associated mucin 4 (MUC4) protein, which has been linked to the increased ability of cancer cells to infiltrate tissues and metastasize. As a result, it has been suggested that miR-150 acts as a tumor suppressor [57]. miRNA-150 has also been linked to the expression of human epidermal growth factor receptor 2 (HER2). Increased levels of miR-150 in PC cells have been shown to lead to a reduction of HER2 receptors, which reduces their ability to invade other tissues and increases cellular adhesion [57].

miRNA-21 is over-expressed in many PC patients. It has been linked to metastases and poor prognosis and may influence the development of an aggressive type of ductal pancreatic adenocarcinoma [61]. miR-21 controls the expression of several important genes, such as silencing *PDCD4*-neoplastic transformation inhibitor and *TIMP3,* both of which act to make PC more aggressive. It also prevents the upregulation of cancer suppressor genes such as *TP53* and cell cycle-dependent kinases, leading to a loss of cell cycle control, the inhibition of apoptosis, and increases in metastasis and disease progression. miRNA-21 expression has been shown to influence the same genes and pathways [58].

## 8. Treatments for Pancreatic Cancer

Current treatment options for pancreatic cancer are limited, highlighting the need for novel therapeutic approaches. Immunotherapy holds promise as a potential breakthrough in pancreatic cancer treatment. Understanding these interactions could lead to innovative treatment strategies and improve patient outcomes. Future research should focus on elucidating the mechanisms underlying these interactions and developing targeted immunotherapies for pancreatic cancer.

Chemotherapy is an important treatment option for pancreatic cancer, either as an adjuvant therapy after surgery or as a primary treatment for advanced disease. Chemotherapy plays a crucial role in the treatment of pancreatic cancer, either as an adjuvant therapy after surgery or as a primary treatment for advanced disease. It can improve survival, relieve symptoms, and improve the quality of life for patients with pancreatic cancer. However, the continued or serial use of chemotherapy, particularly with those drugs that affect tumor cell metabolism and/or signal transduction pathways, has been shown to influence tumor occurrence, tumor recurrence, metastasis, response to the drug, development of drug resistance, and the activation of cancer stem cells (CSCs) [62,63]. The selective pressure of chemotherapy forces the evolution of resistant clones of tumor cells by selecting mutants that can escape the drug. Better, safer, and more targeted treatments remain a major goal for the treatment of PC. 

The unique genetically driven mutations in PC tumor cells pose a significant challenge for molecular targeted therapy. Some examples of targeted therapy drugs used in pancreatic cancer include Erlotinib (Tarceva), which targets EGFR, and Bevacizumab (Avastin), which targets VEGF. Newer-targeted therapies are also being studied in clinical trials.

Targeted therapy has achieved significant success in various types of cancer. For instance, EGFR or VEGF-directed antibodies are effective in colorectal cancer, Trastuzumab is used for HER-2 positive breast cancer, and tyrosine kinase inhibitors like Crizotinib are used for specific subpopulations of non-small cell lung cancer. Despite the recent FDA approval of pembrolizumab as a targeted treatment for PC, other targeted drugs, such as Aflibercept, Cetuximab, Sorafenib, Bevacizumab, and Axitinib, have failed in PC patients. Preliminary evidence suggests that the efficacy of PDAC treatment in patients with high HA expression may depend on PEGPH20 in combination with PARP inhibitors and NAB-Paclitaxel/Gemcitabine [59]. Nanoparticles containing drugs or miRNAs are currently being evaluated for local delivery to PC. This may permit higher local concentrations than could otherwise be achieved while reducing potential side effects. *KRAS* is mutated in about 95% of PC patients. *KRAS* codon 12 mutations occur in about 71% of cases and include G12D (42%), G12V (32%), G12R (15%), G12C (1.5%), G12A (0.4%), and G12S (0.1%) [29]. Since KRAS mutation is a major driver of PC, this is a very active area of development for targeted therapeutics. In phase I/II trials, Sotorasib and Adagrasib showed encouraging results targeting *KRAS* [64]. Another phase I/II trial employing an RNAi targeting *KRAS* G12D also produced encouraging results [65]. A list of ongoing trials targeting *KRAS* and other specific mutations can be seen in Table 3.

Exosomes are extracellular membrane-bound vesicles containing nucleic acids and proteins and are known as mediators of intercellular communication. In PC, exosomes derived from both normal cells and cancer cells play a significant role in the progression of the disease. They can influence PC cell behavior, thereby worsening the prognosis of patients. This can occur via modification of the tumor microenvironment, which, in turn, may increase tumor cell invasiveness and resistance to treatment. On the diagnostic front, exosomes can serve as biomarkers in liquid biopsy for patients with PC, enabling detection before the onset of significant symptoms when therapies may be more effective. With regard to tumor therapeutics, exosomes can be engineered to facilitate more efficient drug delivery to PC tissues. Understanding exosomes has consequently become a major focus in PC research, leading to insights and suggesting avenues for potential future investigations [25].

A clinical study involving 19 patients with metastatic PC evaluated a molecularly tailored treatment regimen. These patients received dual treatment with Gemcitabine and Oxaliplatin, resulting in a reduction of tumor markers by more than 50% in 55% of the patients. Among them, 28% experienced partial remission and 50% had stable disease. However, 88% of the patients still died within three months after treatment, with the remaining patients requiring second-line therapy [66,67]. New targets for PC-targeted therapy, such as PEGPH20 and CKAP4, have recently been identified [68,69].

Over the last decade, there have been major changes in the approach to the development of therapeutics for PC. Next-generation sequencing technology and bioinformatics have resulted in the discovery of driver mutations and altered pathways in PC. Targeted therapies that specifically inhibit key signaling pathways involved in pancreatic cancer have shown promise in preclinical and clinical studies [70]. The commonly targeted pathways include the epidermal growth factor receptor (EGFR), the phosphoinositide 3-kinase *(PI3K)/Akt/mTOR* pathway, the *RAS/RAF/MEK/ERK* pathway, and the Notch signaling pathway. EGFR inhibitors, such as Erlotinib, have been approved for the treatment of pancreatic cancer, but their efficacy is limited due to intrinsic and acquired resistance mechanisms. This topic has recently been reviewed [70] and includes a list of ongoing clinical trials. 

In addition to targeted therapies, several emerging therapeutic strategies are being investigated for the treatment of pancreatic cancer. Immunotherapy, particularly immune checkpoint inhibitors targeting PD-1/PD-L1 and CTLA-4, has shown promise in clinical trials. Other novel approaches, such as oncolytic viruses, cancer vaccines, and epigenetic therapies, are also being explored for their potential efficacy in pancreatic cancer treatment.

A potentially exciting development is the ability of exogenously added miR-34a and miR-143/145, which are normally reduced in PC, to mute the expression of *KRAS.* Adding miR-34a led to improvements in the function of TP53 [59]. This may provide the foundation for a strategy that can supplement or replace traditional therapies. 

Targeted therapy is an important treatment option for pancreatic cancer, either alone or in combination with other treatments. It can improve outcomes for patients with pancreatic cancer and is an area of active research and development. Personalized medicine approaches, which tailor treatment based on the individual characteristics of the tumor, are being explored in pancreatic cancer. This includes identifying specific genetic mutations or biomarkers that can predict treatment response and guide therapy selection.

Future treatment directions for pancreatic cancer are focused on improving early detection, developing targeted therapies, advancing immunotherapy, and personalizing treatment approaches. These efforts aim to improve outcomes and quality of life for patients with pancreatic cancer, which remains one of the most challenging cancers to treat.

## 9. Conclusions

Pancreatic cancer remains a formidable challenge, with limited treatment options and poor prognosis. However, advances in our understanding of the molecular mechanisms driving pancreatic cancer have led to the identification of promising molecular targets for diagnosis and treatment. Further research is needed to validate these targets and develop effective therapies that can improve outcomes for pancreatic cancer patients. 

Due to the considerable variation in the genetic backgrounds of PC, it has become increasingly important to genotype the tumor with the objective of targeting the involved genes. Standard chemotherapeutic treatments are unable to significantly improve survival in PC cases for which surgical intervention is not an option. An example of the importance of evaluating the genetic background can be seen in the case of *KRAS*. Mutant *KRAS* and wild-type *KRAS* tumors are very different tumors, despite histological similarities. Tumors with mutant *KRAS* would benefit from *KRAS* inhibitors, while PC containing wild-type RAS would not respond to those drugs but may benefit from EGFR inhibitors. Antibody–drug conjugates have provided another source of new treatments for PC.

A few new drugs have enabled personalized PC treatment within a limited scope, and new pharmaceuticals are currently being developed and tested in clinical trials. Exosomes and circulating PC cells contain genetic signatures that could enable earlier diagnoses of PC, with earlier intervention hopefully leading to improved outcomes. 

## Figures and Tables

**Figure 1 ijms-25-10843-f001:**
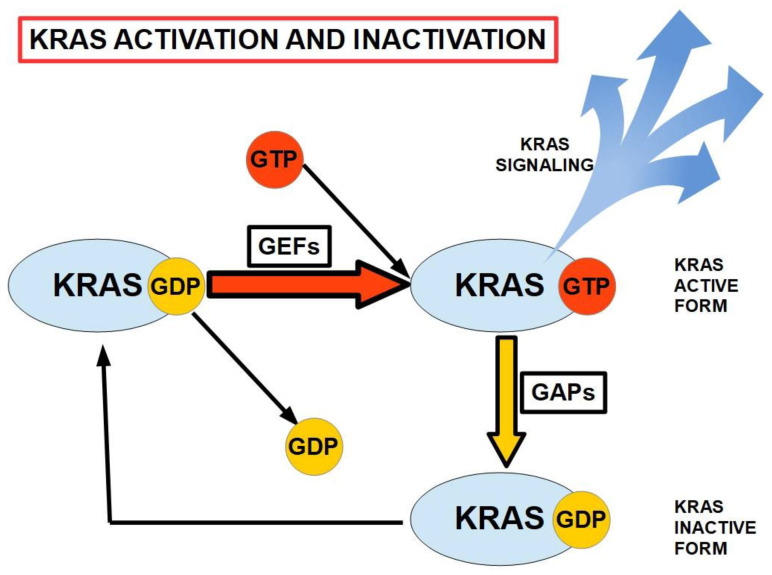
The activation/inactivation cycle for *KRAS.* GEFs and GAPs play major roles in the cyclic binding and hydrolysis of GTP and the removal of GDP from the inactive *KRAS.* GEFs: guanine nucleotide exchange factors. GAPs: GTPase-activating proteins. *KRAS* is activated by GTP binding (adapted from ref. [26]).

**Table 1 ijms-25-10843-t001:** Hypomethylated genes in PC.

Gene	Effects
*CLDN4*	Impedes drug penetration into cells. Increases chemosensitivity.
*LCN2*	Secretes glycoprotein that induces proliferation, angiogenesis, and invasion.
*MSLN*	Interacts with MUC18-favoring peritoneal metastasis and plays a role in drug resistance.
*PSCA*	A surface antigen correlated with advanced stages and tumor progression.
	Over-expressed in metastasis.
*S100A4*	Increases tumor progression and promotes metastasis.
	Inhibits apoptosis and promotes chemoresistance.
*SFN*	Involved in tumor progression, considered a marker of poor prognosis.
*TFF2*	Induces PC cell migration. Also shows anti-tumoral effects.

**Table 2 ijms-25-10843-t002:** Abnormal expression of miRNAs in pancreatic cancer [56].

Circulating miRNAs	Expression Level
miR-21	Increased
miR-155	Increased
miR-196a	Increased
miR-210	Increased
miR-16	Increased
miR-21	Increased
miR-155	Increased
miR181a	Increased
miR-181b	Increased
miR-196a	Increased
miR-210	Increased
miR-26b	Increased
miR-34a	Increased
miR-122	Increased
miR-126	Increased
miR-145	Increased
miR-150	Increased
miR-196a	Increased
miR-223	Increased
miR-505	Decreased
miR-636	Decreased
miR-885.5p	Decreased
miR-18a	Decreased
miR-21	Decreased
miR-221	Decreased
miR-483-3p	Decreased
miR-20a	Decreased
miR-21	Decreased
miR-24	Decreased
miR-25	Decreased
miR-99a	Decreased
miR-185	Decreased
miR-191	Decreased
miR-1246	Decreased
miR-4644	Decreased
miR-3976	Decreased
miR-4306	Decreased
let-7d	Decreased

**Table 3 ijms-25-10843-t003:** Phase I/II clinical trials of targeted therapies for pancreatic cancer.

Target	Drug	Trial Phase	NCT Number
*KRAS*	TCR-T (G12V)	I	NCT04146298
	Exosomes (G12D)	I	NCT03608631
	LY 3537982 (G12C) *	I	NCT04956640
	Anti-KRAS G12V mTCR PBL	I	NCT03190941
	Anti-KRAS G12D mTCR PBL	I	NCT03745326
	MRT X849 **	II	NCT03785249
	ELI-002 ***	I	NCT04853017
*CDKN2A*	SY 5609 (CDK7)	I	NCT04247126
	Palbociclib (CDK 4/6) ***	I	NCT03065062
*BRCA 1 or 2*	Olaparib + Pembrolizumab		
	vs. Olaparib alone	II	NCT04548752
*BRCA1, 2, PALB2*	Adjuvant Olaparib		
	vs. placebo (APOLLO trial)	II	NCT04858334
*ATM*	ATR inhibitors		
*PIK3CA*	Taselisib (MAT CH)	II	NCT02465060
*ROS1*	Crizotinib (MAT CH)	II	NCT02465060
*BRAF*	Dabrafenib/Trametinib	II	NCT02465060
*ERBB2*	Trastuzumab/Deruxtecan	II	NCT04482309
*CTNNB1*	Tegavivint	II	NCT04851119
*NF1*	Trametinib (MAT CH)	II	NCT02465060
*FGFR1*	Trametinib (MAT CH)	II	NCT02465060
*TP53*	SGT53 + GEM + NabP	II	NCT02340117

* in combination with Abemaciclib, Erlotinb, Pembrolizumab, Temuterkib, LY 3295668, Cetuximab, TNO155. ** combined with Pembrolizumab, Cetuximab, Afatinib. *** ELI-002 2P (Amph-modified KRAS peptides, Amph-G12D and Amph-G12R mixed with Amph-CpG-7909) will be evaluated and will transition to the ELI-002 7 P drug formulation containing all seven Amph peptides (G12D, G12R, G12V, G12A, G12C, G12S, G13D) in future clinical trials.

## Data Availability

Not applicable.

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
