# Peer review of "Molecular Targets for the Diagnosis and Treatment of Pancreatic Cancer"

_ijms, 2024, doi:10.3390/ijms251910843_

Round 1

Reviewer 1 Report

Comments and Suggestions for Authors

This is a comprehensive review of molecular targets used for diagnosis and treatment decisions in pancreatic cancer patients. The outline of the article is clear, references are uptodate.

Major comments:

- I missed a section on molecular targets associated with the immunological mechanisms of the antitumor response. Possibilities to discuss this would be a section on immunogenicity and antigenicity of pancreatic cancer versus other tumor entities. Literature on the MSI status of pancreatic cancer patients and resulting consequences for therapy with checkpoint inhibitors should be included.

- PD-1/PDL-1, CTLA-4 and TIM-3 are molecular targets, too, and many researchers interested in these will be guided to this article. I would expect at least one full paragraph on those molecules, including a literature update for pancreatic cancer and immunotherapy. Otherwise the title of the article should be changed.

Minor comments:

- There are some formatting errors (e.g. space before full stop).

- "such as gallstones an cirrhosis" (line 279): "and"

Comments on the Quality of English Language

no comments beyond those on the small formatting errors above.

Author Response

Response to Reviewer 1 Comments

1. Summary

2. Questions for General Evaluation

NA

Reviewer’s Evaluation

NA

Response and Revisions

NA

3. Point-by-point response to Comments and Suggestions for Authors

Comments : This is a comprehensive review of molecular targets used for diagnosis and treatment decisions in pancreatic cancer patients. The outline of the article is clear, references are uptodate.

Major comments:

1)  I missed a section on molecular targets associated with the immunological mechanisms of the antitumor response. Possibilities to discuss this would be a section on immunogenicity and antigenicity of pancreatic cancer versus other tumor entities. Literature on the MSI status of pancreatic cancer patients and resulting consequences for therapy with checkpoint inhibitors should be included.

Response 1: Thank you for pointing this out. We agree with this comment. Therefore, we have added a section on the immunological mechanisms of the antitumor response, and provided references in support of the statements. See Section 5, lines 249 to 280

Comments 2: 2) PD-1/PDL-1, CTLA-4 and TIM-3 are molecular targets, too, and many researchers interested in these will be guided to this article. I would expect at least one full paragraph on those molecules, including a literature update for pancreatic cancer and immunotherapy. Otherwise the title of the article should be changed.

Response 2:  Thank you for pointing this out. Your suggestions have been included in the revised text.  See lines 272 to 280.

4. Response to Comments on the Quality of English Language

Minor comments:

- There are some formatting errors (e.g. space before full stop).

- "such as gallstones an cirrhosis" (line 279): "and"

Point 1: no comments beyond those on the small formatting errors above.

Response 1:    The Errors that were pointed out by the reviewer have been corrected.

Reviewer 2 Report

Comments and Suggestions for Authors

1. Throughout the manuscript, some of the statements require citations, but they appear in two to three sentences afterward.

2. The manuscript has over 232 grammar errors with 6% plagiarism.

3. The authors need to articulate the following clearly: (a) At the point of diagnosis, the PC disease has metastasized and very few treatment options are available to patients. (b) The aggressive nature of this cancer can be attributed to the substantial number of mutations acquired during its progression and its subsequent resistance to standard therapies such as chemotherapy and radiation. (c) The heterogeneity of the chemoresistant subpopulations, particularly the tumor-initiating population as known as cancer stem cells, makes administrating conventional first-line treatments such as gemcitabine more difficult. (d)  Lastly, the tumor microenvironment and the early establishment of a metastatic niche by exosomes that facilitate the dissemination of cancer cells to distant organs also contribute to the incurability of this type of cancer.

4. The manuscript describes some of the current therapies but must also articulate the limitations and adverse side effects.

5. What are the ongoing clinical challenges for drug development? For example, (a) acquired gene mutations and resistance to therapy, (b) implications of pancreatic intratumoral heterogeneity, the presence of this mutation in over 90% of pancreatic cancers, and studies demonstrating
its detection in late-stage cancers, its value as a prognostic tool
is diminished. Despite its ineffective as a prognostic tool, when in addition to CDKN2A, TP53, BRCA2, and SMAD4/DPC4 mutations, they represent key genetic events that are required for PDAC to progress into an aggressive malignancy. These key driver mutations are consistent across the
majority of PDAC, erroneously providing a homogenous genetic profile; however, subpopulations within the tumor acquire their unique genetic profiles as the architectural arrangement of subpopulations can also vary. (b) cancer stem cells (CSC) present another major obstacle in the treatment of PDAC. These dedifferentiated cells have been shown to maintain long-term tumorigenic potential and can regrow new micrometastases, under the regulation of the surrounding tumor microenvironment (TME). (c) Metastatic disease is difficult to treat with conventional chemotherapeutic methods due to its unique genetic repertoire, size, and location within a tissue relative to
the primary tumor. Therefore, it is important to elucidate the molecular events that mediate the development of metastatic disease. For example, exosomes are extracellular membrane-bound vesicles containing nucleic acids and proteins. Exosomes have been implicated in the establishment of a premetastatic niche in liver metastases of patients with PDAC and have been proposed as early detection tools as they circulate in the bloodstream, which makes them an important area of study. Exosome-mediated development of a metastatic TME is proposed to involve the release of PDAC exosomes containing macrophage migration inhibitor factor (MIF) to preferentially
fuse with Kupffer cells (KC) of the liver. The efficacy of exosomes as mediators of metastasis has important implications on early detection strategies and possible therapeutic targets as they circulate the blood and are known
mediators of intercellular communication. (d) Immune-regulated tumorigenesis is another area of interest. Notably, the dense stroma is characterized by a tumor-promoting immunosuppressive environment.

5. Future studies must not rely on targeting a single oncogenic or mitotic
pathway, but must suppress the multiple stages of tumorigenesis of pancreatic cancer cells and their enabling microenvironment.

6. KRAS mutations circulation in plasma in the form of cell-free DNA could potentially be used as an early diagnostic method for individuals exhibiting symptoms of pancreatic cancer and possibly detect the development
of pancreatic cancer in asymptomatic individuals. One might also consider the efficacy of screening for additional mutations that work in conjunction with KRAS to increase the sensitivity of detection. For example, a small-scale study conducted by Pellegata et al. showed that KRAS and p53 mutations
interacted with each other to establish ductal pancreatic cancer. Mutations in both KRAS and p53 were identified to have taken place in pancreatic cell lines and are suggested to lead to a malignant phenotype. The efficacy and sensitivity of early predictive screening of cancer progression require other gene mutations in addition to KRAS.

Comments on the Quality of English Language

The manuscript has over 232 grammar errors with 6% plagiarism.

Author Response

Response to Reviewer 2 Comments

1. Summary

2. Questions for General Evaluation

NA

Reviewer’s Evaluation

NA

Response and Revisions

NA

3. Point-by-point response to Comments and Suggestions for Authors

Comment 1:  Throughout the manuscript, some of the statements require citations, but they appear in two to three sentences afterward.

Comment 2:. The manuscript has over 232 grammar errors with 6% plagiarism.

Comment 3:. The authors need to articulate the following clearly: (a) At the point of diagnosis, the PC disease has metastasized and very few treatment options are available to patients. (b) The aggressive nature of this cancer can be attributed to the substantial number of mutations acquired during its progression and its subsequent resistance to standard therapies such as chemotherapy and radiation. (c) The heterogeneity of the chemoresistant subpopulations, particularly the tumor-initiating population as known as cancer stem cells, makes administrating conventional first-line treatments such as gemcitabine more difficult. (d)  Lastly, the tumor microenvironment and the early establishment of a metastatic niche by exosomes that facilitate the dissemination of cancer cells to distant organs also contribute to the incurability of this type of cancer.

Comment 4:. The manuscript describes some of the current therapies but must also articulate the limitations and adverse side effects.

Comment 5:. What are the ongoing clinical challenges for drug development? For example, (a) acquired gene mutations and resistance to therapy, (b) implications of pancreatic intratumoral heterogeneity, the presence of this mutation in over 90% of pancreatic cancers, and studies demonstrating its detection in late-stage cancers, its value as a prognostic tool
is diminished. Despite its ineffective as a prognostic tool, when in addition to CDKN2A, TP53, BRCA2, and SMAD4/DPC4 mutations, they represent key genetic events that are required for PDAC to progress into an aggressive malignancy. These key driver mutations are consistent across the
majority of PDAC, erroneously providing a homogenous genetic profile; however, subpopulations within the tumor acquire their unique genetic profiles as the architectural arrangement of subpopulations can also vary. (b) cancer stem cells (CSC) present another major obstacle in the treatment of PDAC. These dedifferentiated cells have been shown to maintain long-term tumorigenic potential and can regrow new micrometastases, under the regulation of the surrounding tumor microenvironment (TME). (c) Metastatic disease is difficult to treat with conventional chemotherapeutic methods due to its unique genetic repertoire, size, and location within a tissue relative to the primary tumor. Therefore, it is important to elucidate the molecular events that mediate the development of metastatic disease. For example, exosomes are extracellular membrane-bound vesicles containing nucleic acids and proteins. Exosomes have been implicated in the establishment of a premetastatic niche in liver metastases of patients with PDAC and have been proposed as early detection tools as they circulate in the bloodstream, which makes them an important area of study. Exosome-mediated development of a metastatic TME is proposed to involve the release of PDAC exosomes containing macrophage migration inhibitor factor (MIF) to preferentially fuse with Kupffer cells (KC) of the liver. The efficacy of exosomes as mediators of metastasis has important implications on early detection strategies and possible therapeutic targets as they circulate the blood and are known mediators of intercellular communication. (d) Immune-regulated tumorigenesis is another area of interest. Notably, the dense stroma is characterized by a tumor-promoting immunosuppressive environment.

Comment 5a:. Future studies must not rely on targeting a single oncogenic or mitotic
pathway, but must suppress the multiple stages of tumorigenesis of pancreatic cancer cells and their enabling microenvironment.

Comment 6:. KRAS mutations circulation in plasma in the form of cell-free DNA could potentially be used as an early diagnostic method for individuals exhibiting symptoms of pancreatic cancer and possibly detect the development of pancreatic cancer in asymptomatic individuals. One might also consider the efficacy of screening for additional mutations that work in conjunction with KRAS to increase the sensitivity of detection. For example, a small-scale study conducted by Pellegata et al. showed that KRAS and p53 mutations interacted with each other to establish ductal pancreatic cancer. Mutations in both KRAS and p53 were identified to have taken place in pancreatic cell lines and are suggested to lead to a malignant phenotype. The efficacy and sensitivity of early predictive screening of cancer progression require other gene mutations in addition to KRAS.

Comment 1:  Throughout the manuscript, some of the statements require citations, but they appear in two to three sentences afterward.

Response 1: Respectfully, we have followed the MLA format for multiple contiguous citations from the same source. We choose to continue to use this format.

Comment 3:. The authors need to articulate the following clearly: (a) At the point of diagnosis, the PC disease has metastasized and very few treatment options are available to patients. (b) The aggressive nature of this cancer can be attributed to the substantial number of mutations acquired during its progression and its subsequent resistance to standard therapies such as chemotherapy and radiation. (c) The heterogeneity of the chemoresistant subpopulations, particularly the tumor-initiating population as known as cancer stem cells, makes administrating conventional first-line treatments such as gemcitabine more difficult. (d)  Lastly, the tumor microenvironment and the early establishment of a metastatic niche by exosomes that facilitate the dissemination of cancer cells to distant organs also contribute to the incurability of this type of cancer.

Response 3: We agree that these are very important points, and we would be very pleased to  add these comments to the text.  Your description is concise and elegant, and we would like to use them verbatim in the text. We would be pleased to credit you as a source, but as you know, we do not know who you are.  If you provide the information, we will add you as a reference.  Otherwise, it will be credited as a “personal communication”

Comment 4:. The manuscript describes some of the current therapies but must also articulate the limitations and adverse side effects

Response 4: We have added lines 373 to 379 to address the limitations of drug treatment. 

Comment 5:. What are the ongoing clinical challenges for drug development? For example, (a) acquired gene mutations and resistance to therapy, (b) implications of pancreatic intratumoral heterogeneity, the presence of this mutation in over 90% of pancreatic cancers, and studies demonstrating its detection in late-stage cancers, its value as a prognostic tool
is diminished. Despite its ineffective as a prognostic tool, when in addition to CDKN2A, TP53, BRCA2, and SMAD4/DPC4 mutations, they represent key genetic events that are required for PDAC to progress into an aggressive malignancy. These key driver mutations are consistent across the
majority of PDAC, erroneously providing a homogenous genetic profile; however, subpopulations within the tumor acquire their unique genetic profiles as the architectural arrangement of subpopulations can also vary. (b) cancer stem cells (CSC) present another major obstacle in the treatment of PDAC. These dedifferentiated cells have been shown to maintain long-term tumorigenic potential and can regrow new micrometastases, under the regulation of the surrounding tumor microenvironment (TME). (c) Metastatic disease is difficult to treat with conventional chemotherapeutic methods due to its unique genetic repertoire, size, and location within a tissue relative to the primary tumor. Therefore, it is important to elucidate the molecular events that mediate the development of metastatic disease. For example, exosomes are extracellular membrane-bound vesicles containing nucleic acids and proteins. Exosomes have been implicated in the establishment of a premetastatic niche in liver metastases of patients with PDAC and have been proposed as early detection tools as they circulate in the bloodstream, which makes them an important area of study. Exosome-mediated development of a metastatic TME is proposed to involve the release of PDAC exosomes containing macrophage migration inhibitor factor (MIF) to preferentially fuse with Kupffer cells (KC) of the liver. The efficacy of exosomes as mediators of metastasis has important implications on early detection strategies and possible therapeutic targets as they circulate the blood and are known mediators of intercellular communication. (d) Immune-regulated tumorigenesis is another area of interest. Notably, the dense stroma is characterized by a tumor-promoting immunosuppressive environment.

Response 5: Thank you for these comments.  We have included information regarding exosomes in PC (lines 404 to 415).  We have added a section (Section 5) on Immunogenicity and Antigenicity of PC (lines 249 to 280). We feel that the issue of driver mutations has been covered sufficiently in the original text. Respectfully, we believe that the mechanism by which metastasis arises in tumors is beyond the scope of this article.

Comment 5a:. Future studies must not rely on targeting a single oncogenic or mitotic
pathway, but must suppress the multiple stages of tumorigenesis of pancreatic cancer cells and their enabling microenvironment.

Response 5a: We agree, and we have once again chosen to use your language, with attribution as a personal communication. (lines 224 to 226)

Comment 6:. KRAS mutations circulation in plasma in the form of cell-free DNA could potentially be used as an early diagnostic method for individuals exhibiting symptoms of pancreatic cancer and possibly detect the development of pancreatic cancer in asymptomatic individuals. One might also consider the efficacy of screening for additional mutations that work in conjunction with KRAS to increase the sensitivity of detection. For example, a small-scale study conducted by Pellegata et al. showed that KRAS and p53 mutations interacted with each other to establish ductal pancreatic cancer. Mutations in both KRAS and p53 were identified to have taken place in pancreatic cell lines and are suggested to lead to a malignant phenotype. The efficacy and sensitivity of early predictive screening of cancer progression require other gene mutations in addition to KRAS.

Response 6:  We thank you for this comment, and have included text to comply (lines 141 to 146.

4. Response to Comments on the Quality of English Language

Comment 2: The manuscript has over 232 grammar errors with 6% plagiarism.

Response 2: This comment was difficult to understand, since we had used a word processor that performs extensive spelling and grammatical checking.  When we realized that the manuscript we uploaded was NOT identical to the one on the website, the reason became clear.  There were some errors that we made, and these were fixed.  However, I am not certain that the same problem will not recur when we upload the revised manuscript.  We hope this is not the case.  We are not aware of what plagiarism to refer, but if you can point it out, will gladly cite the source or remove the offending section.

5. Additional clarifications

Round 2

Reviewer 1 Report

Comments and Suggestions for Authors

All suggestions were addressed.

Reviewer 2 Report

Comments and Suggestions for Authors

Comments on the Quality of English Language